# Improvement of the Bias Stress Stability in 2D MoS_2_ and WS_2_ Transistors with a TiO_2_ Interfacial Layer

**DOI:** 10.3390/nano9081155

**Published:** 2019-08-12

**Authors:** Woojin Park, Yusin Pak, Hye Yeon Jang, Jae Hyeon Nam, Tae Hyeon Kim, Seyoung Oh, Sung Mook Choi, Yonghun Kim, Byungjin Cho

**Affiliations:** 1Department of Advanced Material Engineering, Chungbuk National University, Chungdae-ro 1, Seowon-Gu, Cheongju, Chungbuk 28644, Korea; 2Department of Nanobio Materials and Electronics, GIST, 123 Cheomdan-gwagiro, Buk-gu, Gwangju 61005, Korea; 3Materials Center for Energy Department, Surface Technology Division, Korea Institute of Materials Science (KIMS), 797 Changwondaero, Sungsan-gu, Changwon, Gyeongnam 51508, Korea

**Keywords:** MoS_2_, WS_2_, interfacial layer, contact resistance, bias stress stability

## Abstract

The fermi-level pinning phenomenon, which occurs at the metal–semiconductor interface, not only obstructs the achievement of high-performance field effect transistors (FETs) but also results in poor long-term stability. This paper reports on the improvement in gate-bias stress stability in two-dimensional (2D) transition metal dichalcogenide (TMD) FETs with a titanium dioxide (TiO_2_) interfacial layer inserted between the 2D TMDs (MoS_2_ or WS_2_) and metal electrodes. Compared to the control MoS_2_, the device without the TiO_2_ layer, the TiO_2_ interfacial layer deposited on 2D TMDs could lead to more effective carrier modulation by simply changing the contact metal, thereby improving the performance of the Schottky-barrier-modulated FET device. The TiO_2_ layer could also suppress the Fermi-level pinning phenomenon usually fixed to the metal–semiconductor interface, resulting in an improvement in transistor performance. Especially, the introduction of the TiO_2_ layer contributed to achieving stable device performance. Threshold voltage variation of MoS_2_ and WS_2_ FETs with the TiO_2_ interfacial layer was ~2 V and ~3.6 V, respectively. The theoretical result of the density function theory validated that mid-gap energy states created within the bandgap of 2D MoS_2_ can cause a doping effect. The simple approach of introducing a thin interfacial oxide layer offers a promising way toward the implementation of high-performance 2D TMD-based logic circuits.

## 1. Introduction

The process of extreme scaling-down to reach a physical channel length limit of sub-100 nm has caused critical problems, such as a short channel effect and increased leakage current. To address these limitations, efforts have recently been made to scrutinize promising semiconducting materials. In particular, atomically thin layered transition metal dichalcogenides (TMDs) have attracted great attention due to their extraordinary electrical, optical, and mechanical properties [1,2,3,4,5,6,7,8,9]. One of their most attractive properties is the existence of a band-gap and its facile engineering. For instance, single-layer molybdenum disulfide (MoS_2_) has a direct band-gap of ~1.8 eV, and multilayer MoS_2_ has an indirect band-gap of ~1.2 eV [1]. The physical properties of 2D TMDs have led to their applications in various electronic devices such as transistors, memory devices, and opto-electronic devices [10,11,12,13,14,15,16,17,18]. Among them, the most promising device is the field caused effect transistor (FET), which functions as an essential switching component of display back-plane circuits [12].

However, a few challenging issues around employing 2D TMD-based FETs for practical applications have to be resolved. Fabricating large-scale, high-quality continuous 2D TMD films and the direct deposition of the gate dielectric layer on a 2D surface with a low surface energy are important issues in terms of the utilization of conventional Si fabrication infrastructures and the realization of high-mobility FETs. Furthermore, the unreliable performance of 2D TMD FETs has been a critical concern that must be preferentially addressed. Chemically and mechanically disordered surface and interface states are the origin of the performance instability of semiconductor devices, causing a large hysteresis window and a significant threshold voltage (V_TH_) shift.

The passivation of the polymer layer on the 2D TMDs is an efficient countermeasure against the instability of 2D semiconductor-based FET performance [19,20]. Using a similar method, Zheng et al. reported that the hysteresis window of the 2D layered materials capped with an Al_2_O_3_ was considerably reduced [20]. Meanwhile, the contact engineering strategy for modifying the interface states between a metal and a 2D semiconductor has been actively studied [21,22,23,24,25,26,27,28]. Because the operation of the 2D TMD FET is based on a modulation of the Schottky-barrier, the interface quality at the metal/TMD contact becomes more critical. Several approaches to reduce the contact resistance, including a doping technique and selection of proper work function metal, have been proposed [26,28]. Meanwhile, Fermi-level pinning usually occurs at a metal/semiconductor contact region, causing high contact resistance due to a fixed high band offset regardless of the work function value of the metal [25,26,29]. Because the interface states usually serve as carrier trapping sites, it is hard to realize the high performance of 2D TMD FETs. Thus, a reliable and simple approach for Fermi-level depinning is necessary. The corresponding result was reported for an MoS_2_ device with an interfacial oxide layer [29].

Herein, the effect of the interfacial buffer layer at the metal/2D TMD (MoS_2_ and WS_2_) contact on transistor performance was experimentally and theoretically investigated. Titanium dioxide (TiO_2_) was used as a buffer layer because its band offset with MoS_2_ and WS_2_ is relatively small and tunnel resistance can be minimized with the thin TiO_2_ layer. By employing a TiO_2_ interlayer, interface states were successfully reduced, achieving an increased drive current and the enhancement of long term bias stress stability. In addition, the role of the TiO_2_ layer on MoS_2_ was theoretically elucidated using a density function theory (DFT) simulation. It can be highlighted that we suggested a facile approach to achieve both higher transistor performance and stability at the same time.

## 2. Materials and Methods

A mechanical exfoliation method using scotch tape to obtain high-quality 2D TMD flakes was adopted, and then the exfoliated 2D TMD flakes (MoS_2_ and WS_2_) were transferred onto a SiO_2_ (300 nm)/heavily doped Si substrate. To identify the existence of the 2D TMDs, MoS_2_ was mechanically exfoliated from the bulk mineral, and the multilayer MoS_2_ was characterized using Raman spectroscopy (Figure 1a). LabRAM ARAMIS (laser wavelength: 473 nm, 50 mW) was used for Raman measurements. Two prominent peaks feature the in-plane E^1^_2g_ mode (~384 cm^−1^) and the out-of-plane A_1g_ mode (~409 cm^−1^) of the MoS_2_. A frequency difference of ~25 cm^−1^ between two vibrational modes indicates a multilayer MoS_2_. To determine the thickness of the exfoliated MoS_2_, we performed an atomic force microscopy (AFM) analysis. As shown in Figure 1b, the 92 nm-thick MoS_2_ was transferred onto the SiO_2_/Si substrate using a typical scotch-tape exfoliation method.

To investigate the effect of the TiO_2_ interlayer on the device’s contact properties, 2D FET devices with back gate electrodes were fabricated: a control device without TiO_2_ and a testing device with TiO_2_. Figure 1c shows the 3D schematic image of the FET device with the 2D TMD-TiO_2_-Ti/Au structure. The TiO_2_ interfacial layer on the 2D TMDs was deposited using an atomic layer deposition (ALD) technique based on a tetrakis-dimethyl-amido-titanium (TDMAT) precursor at 200 °C. The pulse and purging times were 0.2 s and 20 s, respectively. The number of cycles were 15, resulting in a 2~3 nm thickness. The thickness of the TiO_2_ layer was also optimized to avoid high tunnel resistance. The 2D TMD transistor devices were made by a conventional photolithography process. Photolithography was conducted after spin-coating of the photoresist (AZ 5214, MicroChemicals, Germany), and the metal was deposited by a physical vapor evaporator. Electron beam evaporation was selected to minimize the physical damage on the surface of the TMDs. Lift-off processes were sequentially performed to make the source and drain electrodes. The channel distance between source and drain was ~3 μm. After device fabrication, the post-annealing process was conducted in a vacuum environment at 300 °C. The process of the vacuum annealing step included a 30 min ramping time to 300 °C, for a 1 h duration, and a cool down at room temperature. The electrical characterization (transfer, output, and stress measurement) was performed with a Keithley 4200-SCS (Keithley, Cleveland, OH, US). Stress measurement followed the conventional stress-measure-stress sequence for 10,000 s, which is summarized in Appendix A information.

Figure 1d shows a cross-sectional high-resolution transmission electron microscopy (HRTEM) image of the MoS_2_-TiO_2_-Ti stacked structure. The lattice constant of the MoS_2_ was measured to be ~0.65 nm along the c-plane [0001] direction in a hexagonal close-packed crystal structure. A thin (~3 nm-thick) TiO2 layer, deposited using the atomic layer deposition process, was inserted between the Ti metal and MoS_2_. Interestingly, the discontinuous layers of the MoS_2_ layers exhibited a step-like crystal structure. Thus, it is reasonably expected that randomness in the defect density for the exposed edge planes and basal planes can cause considerable deviation from the physical interface states, thereby inducing a large difference in the electrical properties of MoS_2_. The structural disorder of the MoS_2_ surface is also a strong source for Fermi-level pinning, which caused some points of the band gap to be locked (pinned) to the Fermi-level. This made the Schottky-barrier height considerably insensitive to the metal’s work function. The Fermi-level pinning phenomenon, with respect to various metals (for instance, Ti, Cr, Au, and Pd), is illustrated in Appendix A information. Even in the corresponding literature studies, the existence of dangling bonds in TMD has been proven via in-depth analyses, such as scanning tunneling microscopy and inductively coupled plasma-mass spectroscopy [30,31,32,33].

## 3. Results and Discussion

To investigate the influence of a TiO_2_ interfacial layer on the MoS_2_ and WS_2_ device performance, electrical measurements were performed. Basic electrical characterizations were carried out with a Keithley 4200-SCS (Keithley, Cleveland, OH, US) analyzer. Figure 2a shows a comparison between the transfer characteristics (I_DS_-V_BG_) of the MoS_2_-Ti and MoS_2_-TiO_2_-Ti devices. The gate-bias sweeping ranged from −50 to 20 V at a fixed drain voltage of 0.1 V. A typical unipolar n-type behavior and a depletion mode of MoS_2_ transistor devices were observed. The MoS_2_-TiO_2_-Ti device with a TiO_2_ interfacial layer showed more enhanced performance with a higher drive on current (I_ON_). I_ON_ values for devices without and with the TiO_2_ layer are 0.36 and 1.22 µA, respectively. The field effect mobility (μ_FE_) values for MoS_2_-Ti and MoS_2_-TiO_2_-Ti devices were estimated to be 1.38 and 6.08 cm^2^/V·s, respectively. The transfer curves at variable drain voltages and output characteristic also confirmed the better performance of the testing devices with the TiO_2_ layer (Appendix A information). The μ_FE_ values of the MoS_2_-TiO_2_-Ti device as a function of gate voltage were higher than those of the MoS_2_-Ti device (Appendix A information).

A more interesting result was observed on the WS_2_ FETs. Figure 2b shows a comparison of the I_DS_-V_BG_ transfer characteristics of the WS_2_-Ti and WS_2_-TiO_2_-Ti structured devices. The bi-polar behavior of the WS_2_-Ti structured devices was observed, which is consistent with the previous results [34]. It is highly likely that the Fermi-level of the Ti metal exists within the mid-gap of WS_2_. The transfer curve of the WS_2_-TiO_2_-Ti structured device showed stronger n-type unipolar behavior with a higher I_ON_ current than that of the WS_2_-Ti device. As shown in Figure 2c, we also characterized the WS_2_ devices using Pd metal electrodes with a relatively high work function of ~5.1 eV to understand the mid-gap pinning and the effects of an interfacial layer. The addition of the TiO_2_ layer on the WS_2_ caused a change from a weak bipolar to a p-type unipolar behavior. This result indicates that a high Schottky-barrier can be effectively reduced by a contact engineering approach utilizing a very thin TiO_2_ interfacial layer. The I_DS_-V_BG_ curves of the WS_2_ FETs at various drain voltages are also shown in Appendix A information. The performance enhancement of the 2D FET devices with the interfacial TiO_2_ layer is attributed to the considerable reduction in the density of the diverse interface states, resulting from the direct contact between the metal and the 2D semiconductor channel. Comparison of the proposed band diagrams between the 2D TMD-Ti and 2D TMD-TiO_2_-Ti devices highlights the change in the Schottky-barrier height as shown in Appendix A information. In principle, the theoretical Fermi-level alignment between the metal and semiconductor, called Fermi-level depinning, also creates a more effective carrier modulation of the 2D TMD FET device.

For practical transistor applications, the electrical stability of the MoS_2_ based FET devices was examined under a long-term positive gate-bias stress condition, as shown in Figure 3a–d. Figure 3a,b shows the shift of the I_DS_-V_BG_ curves during the long-term gate-bias stress test. The transfer I-V curve properties were monitored every logarithmic time interval (1, 10, 100, 1000, and 10,000 s) while continuously applying +10 V to the gate electrode. Schemes to illustrate the stress measurement set up environment and the data checking points are shown in Appendix A information. Even if the I_DS_-V_BG_ curves in all of the MoS_2_-Ti and MoS_2_-TiO_2_-Ti devices were slightly shifted to the positive direction, the device with the TiO_2_ layer showed less of a shift than that without TiO_2_, indicating more stable electrical properties compared to the control device without TiO_2_. Interestingly, in Figure 3b, the variation of I_OFF_ values for the MoS_2_-TiO_2_-Ti stack seems more severe than that of the control MoS_2_-Ti device. The actual differences of the minimum and maximum I_OFF_ values are 4.20 × 10^−12^ A and 3.68 × 10^−10^ A for MoS_2_-Ti and MoS_2_-TiO_2_-Ti, respectively. The I_OFF_ fluctuation of all the devices was less than 1 nA, and this fluctuation was negligible in operation. Figure 3c shows a summary of the threshold voltage (V_TH_) change for MoS_2_-Ti and MoS_2_-TiO_2_-Ti stacked devices as a function of stress time, which was extracted from the raw data from Figure 3a,b. The MoS_2_ FET without a TiO_2_ layer showed a more positive V_TH_ shift than that of the MoS_2_ FET with a TiO_2_ layer. The V_TH_ shift for the MoS_2_ FET without and with a TiO_2_ interfacial layer was 3.1 and 1.1 V, respectively. The TiO_2_ layer could serve as a buffer layer to mitigate the interfacial damage from electrical stress. As shown in Figure 3d, we also compared the field-effect mobility (μ_FE_) values for the devices without and with a TiO_2_ layer. The μ_FE_ was estimated by following equation:
$$\mu_{FE}=g_{m}\frac{L}{W}\frac{1}{V_{DS}}\frac{1}{C_{ox}}\;\mathrm{and}\;g_{m}=\frac{\partial I_{D}}{\partial V_{G}}$$
where *g_m_* is the maximum transconductance that can be achieved from I_DS_-V_BG_, *L* is the channel length, *W* is the channel width, *V_DS_* is the applied drain bias, and *C_ox_* is the gate oxide capacitance.

Overall, the μ_FE_ of MoS_2_-TiO_2_-Ti device was higher than that of the MoS_2_-Ti device. After 10,000 s stress time, the μ_FE_ was reduced from 0.22 to 0.17 cm^2^/Vs for the device without a TiO_2_ layer and from 8.05 to 6.14 cm^2^/Vs for the device with a TiO_2_ layer. Approximately, 25% of the μ_FE_ reduction was observed for both cases.

Additionally, the stability of the contact region for the WS_2_-based FET devices was also determined for the effect of the interfacial TiO_2_ layer on bias stress stability, as shown in Figure 4a,b. As can be seen, the transfer curves of the WS_2_-Ti contact FET device showed bipolar behavior where both electron and hole carriers contribute to the current flow of the semiconductor channel. Overall, a lower V_TH_ shift was observed for the FET with a TiO_2_ layer compared to the FET without a TiO_2_ layer, indicating that the introduction of the TiO_2_ interfacial layer on the WS_2_ layered film is also an effective approach for improving the contact reliability of the WS_2_ device, as well as the case of MoS_2_ device. Specifically, the V_TH_ shifts for the WS_2_ FET without and with a TiO_2_ interfacial layer were 8 and 4.3 V, respectively (Figure 4c). As shown in Figure 4d, the change of μ_FE_ as a function of stress time was also fitted: the mobility value was almost unchanged for the control device without a TiO_2_ layer and from 0.41 to 0.18 cm^2^/Vs for the testing device with a TiO_2_ layer.

Indeed, the WS_2_ FET device was more vulnerable to electrical stress than MoS_2_, which might be due to greater number of interface states at the metal/semiconductor contact. The metal-induced gap states indispensably exist on the metal/semiconductor interface, which induces the instability of transistor performance. Additionally, there is a quantum mechanically long distance of 2–3 Å between the metal and 2D TMD, which increases the tunneling probability of charge carriers [35]. The more stable performance of the 2D TMD devices with an insulating TiO_2_ layer might be understood by a mitigation of those gap states and a reduction in physical distance.

To unveil how the TiO_2_ layer electronically influences the MoS_2_ semiconductor, we explored a theoretical simulation of electronic states for free-standing MoS_2_ and MoS_2_/TiO_2_ materials via a density functional theory (DFT) calculation (Figure 5). The density of states (DOS) calculation result of the free standing MoS_2_ showed the existence of a forbidden gap (Figure 5a). Meanwhile, the TiO_2_/MoS_2_ hybrid combination featured a spin-polarized metallic behavior. The calculated DOS clearly validates that the addition of the TiO_2_ layer leads to the modification of the electronical band structure of the junction region, offering the benefit of a doping effect on MoS_2_.

## 4. Conclusions

The effect of a TiO_2_ interfacial layer on metal/TMD (MoS_2_ and WS_2_) contact was experimentally and theoretically studied. The advantages of a Schottky-type FET device, possibly implemented according to the value of a metal work function, were achieved in the 2D TMD devices with a TiO_2_ layer. Furthermore, a more enhanced and stable electrical performance for the 2D TMD FET devices with the TiO_2_ interfacial layer could be obtained under a gate-bias stress condition. The TiO_2_ interfacial layer could serve as a Fermi-level de-pinning layer, reducing the density of the interface states. Additionally, the DFT calculation validates the doping effect of the TiO_2_ interfacial layer on the 2D MoS_2_. The strategy of inserting a very thin insulating layer into the contact region will be also applied to diverse 2D TMD-based FET devices.

## Figures and Tables

**Figure 1 nanomaterials-09-01155-f001:**
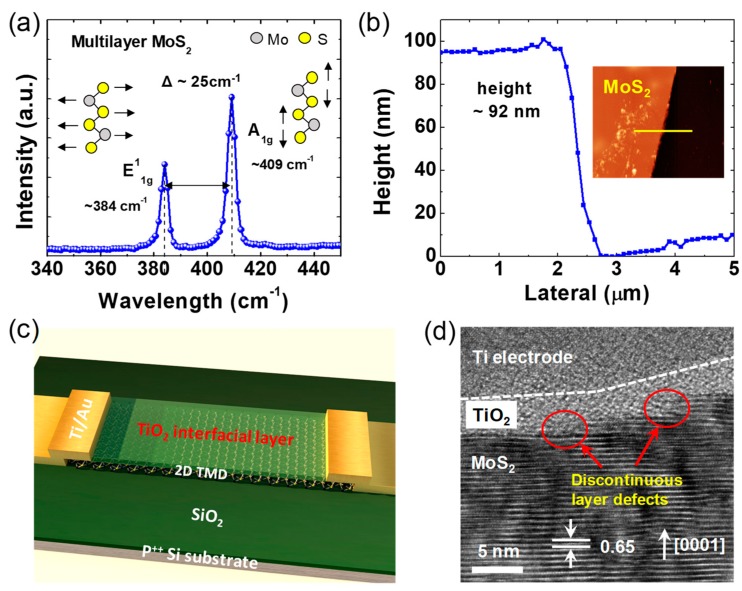
(**a**) Raman spectrum and (**b**) atomic force microscopy (AFM) analysis of a multilayer MoS_2_; (**c**) 3D schematic image of a transition metal dichalcogenide field effect transistor (TMD FET) device; (**d**) a high-resolution transmission electron microscopy (HRTEM) image of the MoS_2_-TiO_2_-Ti stacked structure.

**Figure 2 nanomaterials-09-01155-f002:**
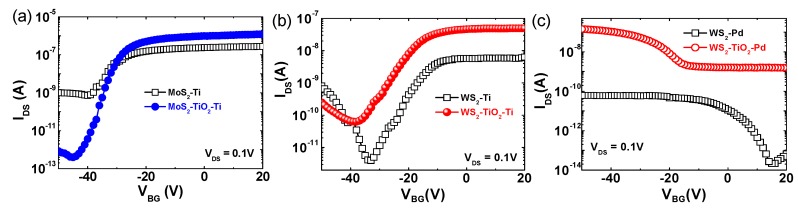
Transfer curves (I_DS_-V_BG_) for (**a**) MoS_2_-Ti and MoS_2_-TiO_2_-Ti, (**b**) WS_2_-Ti and WS_2_-TiO_2_-Ti, (**c**) and WS_2_-Pd and WS_2_-TiO_2_-Pd.

**Figure 3 nanomaterials-09-01155-f003:**
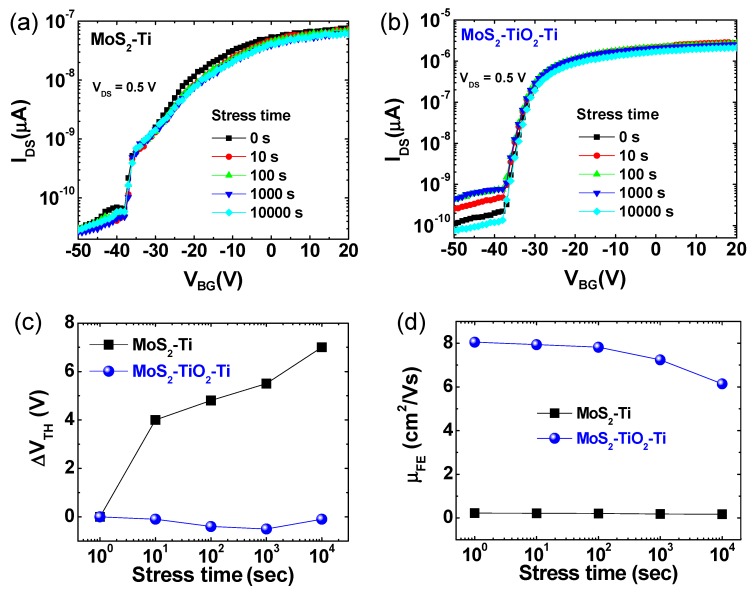
Transfer curves (I_DS_-V_BG_) of MoS_2_ FETs (**a**) without TiO_2_ and (**b**) with a TiO_2_ layer during a 10,000 s gate-bias stress measurement at room temperature. The summary of (**c**) the ΔV_TH_ shift and (**d**) the μ_FE_ change as function of stress time for MoS_2_-Ti and MoS_2_-TiO_2_-Ti.

**Figure 4 nanomaterials-09-01155-f004:**
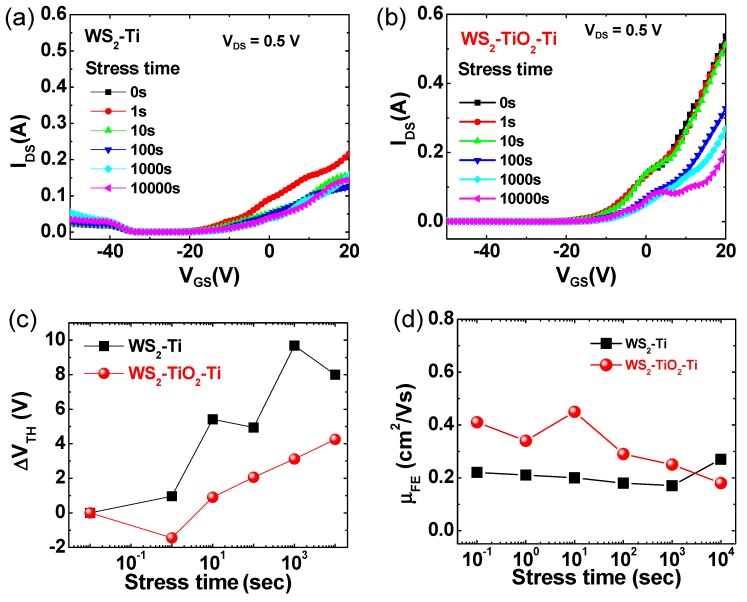
Transfer curves (I_DS_-V_BG_) of WS_2_ FETs (**a**) without TiO_2_ and (**b**) with a TiO_2_ layer during a 10,000 s gate-bias stress measurement at room temperature. The summary of (**c**) the ΔV_TH_ shift and (**d**) the μ_FE_ change as a function of the stress time for WS_2_-Ti and WS_2_-TiO_2_-Ti devices.

**Figure 5 nanomaterials-09-01155-f005:**
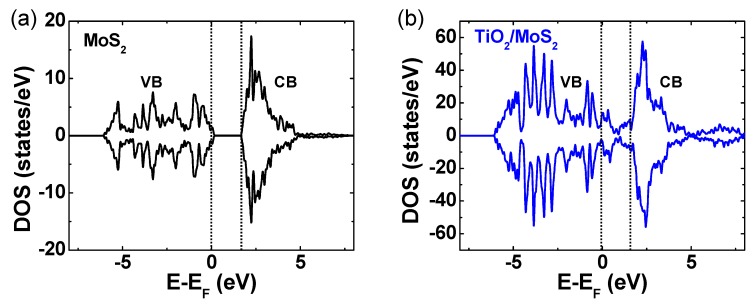
Density function theory (DFT)-calculated density of states (DOS) of (**a**) MoS_2_ and (**b**) TiO_2_/MoS_2._

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
