# Peer review of "Improvement of the Bias Stress Stability in 2D MoS2 and WS2 Transistors with a TiO2 Interfacial Layer"

_nanomaterials, 2019, doi:10.3390/nano9081155_

Round 1

Reviewer 1 Report

The manuscript nanomaterials-559878 presents the improvement of bias stress stability in 2D MoS2 and 2 WS2 transistors with TiO2 interfacial layer.

In the abstract some quantitative results regarding the developed device should be included.

In the Introduction should be better highlighted the advantages of these devices comparing with the similar ones must be included. It is necessary to be evidenced the progress beyond the state of the art.

The details regarding the equipments used for the development of devices, the sources of materials, the apparatus used for the characterization of the devices must be included in  the Materials and Method section.

Results section must be named results and Discussions

The field effect mobility estimation should be included in the manuscript.

Figure 3 a and b. Why for the MoS2 FETs with TiO2 layer (b) the IDS values  in the initial part are different I function of the stress time and in the case of the MoS2 FETs (a) the differences are not visible?

Figure 3 c and d. There are linear fittings? If there are the equations and R2 must be included. The linearity is not clear in all the cases.

Figure 4 a and b. The signals of the developed device are worse comparing with the reference one. Please explain.

Figure 4 c. The linearity in both situations is overestimated. Please reformulate your conclusions and discuss the results in an objective way.

Figure 4 d. The difference between first and second point is in the logarithmic range. The straight line between these points is also overestimated. Please prove this by more experimental points in this range.

Author Response

Please find the revision file attached.

Reviewer 2 Report

In this article the authors investigate the role of an interfacial TiO2 layer prepared by ALD in 2D MoS2 and WS2 field effect transistors. The authors show the interfacial layer improves electrical performance and stability and, further, support a hypothesis to the role of the TiO2 layer with DFT calculations. Overall the authors provided a compelling manuscript that would be of interest to the readership of this journal, however, a few comments should be addressed prior to publication.

1) The experimental section is too vague. While the main processing steps are presented, there is insufficient detail for others in the field to reproduce their work. Please include information as to how the TMD was transferred from the Scotch tape to the Si, expand upon the ALD section (number of cycles, oxidant, pulse and purge times), and duration of vacuum anneal step. 

2) Figure 1b&d provide height and interface of the MoS2/TiO2. A table of the thicknesses for all the devices would be useful, especially for comparing the control vs the test devices. Include a TEM image of the WS2 film in the SI; does it have similar step-like structure?

3) There appears to be a significant amount of scatter in the data for the delta-V shift for the WS2-Ti in Figure 4c compared with all other devices presented in the manuscript. Can you provide any reasoning for this scatter? Were multiple WS2-Ti devices tested to see if the scatter was consistent or if there was perhaps a contact issue?

Author Response

Please find the revision file attached.

Round 2

Reviewer 1 Report

 The responses of authors towards referee commentaries and questions are good enough and explain well the unclear parts of the manuscript. The manuscript was carefully revised by the authors and it was improved and completed. Therefore, I recommend the publication of the manuscript without further modifications.